# Genome editing abrogates angiogenesis in vivo

Xionggao Huang[1,2,3], Guohong Zhou[1,2,4], Wenyi Wu[1,2,5], Yajian Duan[1,2,4], Gaoen Ma[1,2], Jingyuan Song[1,2,6], Ru Xiao[1,2], Luk Vandenberghe[1,2], Feng Zhang[7], Patricia A. D'Amore[1,2] & Hetian Lei[1,2]

Angiogenesis, in which vascular endothelial growth factor receptor (VEGFR) 2 plays an essential role, is associated with a variety of human diseases including proliferative diabetic retinopathy and wet age-related macular degeneration. Here we report that a system of adeno-associated virus (AAV)-mediated clustered regularly interspaced short palindromic repeats (CRISPR)-associated endonuclease (Cas)9 from *Streptococcus pyogenes* (SpCas9) is used to deplete VEGFR2 in vascular endothelial cells (ECs), whereby the expression of SpCas9 is driven by an endothelial-specific promoter of intercellular adhesion molecule 2. We further show that recombinant AAV serotype 1 (rAAV1) transduces ECs of pathologic vessels, and that editing of genomic *VEGFR2* locus using rAAV1-mediated CRISPR/Cas9 abrogates angiogenesis in the mouse models of oxygen-induced retinopathy and laser-induced choroid neovascularization. This work establishes a strong foundation for genome editing as a strategy to treat angiogenesis-associated diseases.

[1] Schepens Eye Research Institute of Massachusetts Eye and Ear, Harvard Medical School, Boston, MA 02114, USA. [2] Department of Ophthalmology, Harvard Medical School, Boston, MA 02114, USA. [3] Hainan Eye Hospital, Haikou, Hainan Province 570311, China. [4] Shanxi Eye Hospital, Taiyuan, Shanxi Province 030002, China. [5] Department of Ophthalmology, Second Xiangya Hospital, Second Xiangya Hospital, Central South University, Changsha, Hunan Province 410013, China. [6] Institute of Medicinal Plant Development, Chinese Academy of Medical Sciences & Peking Union Medical College, Beijing 100193, China. [7] Broad Institute of the Massachusetts Institute of Technology and Harvard University, Cambridge, MA 02142, USA. Xionggao Huang, Guohong Zhou, and Wenyi Wu contributed equally to this work. Correspondence and requests for materials should be addressed to H.L. (email: hetian_lei@meei.harvard.edu)

Vascular endothelial growth factor (VEGF) plays a critical role in angiogenesis, the process by which new blood vessels grow from pre-existing vessels[1–3]. Among the VEGF receptors 1, 2, and 3 (VEGFR1, 2, and 3), VEGFR2 mediates nearly all known VEGF-induced output, including microvascular permeability and neovascularization (NV)[4]. NV is critical for supporting the rapid growth of solid tumors beyond $1–2 mm^3$ and for tumor metastasis[5]. Abnormal angiogenesis is also associated with a variety of other human diseases such as proliferative diabetic retinopathy (PDR)[6, 7], retinopathy of pre-maturity (ROP)[8], and wet age-related macular degeneration (AMD)[9, 10]. PDR accounts for the highest incidence of acquired blindness in the working age population[6, 7]; ROP is a major cause of acquired blindness in children[8]; AMD represents the leading cause of blindness in people over the age of 65 afflicting 30–50 million people globally[10]. Preventing VEGF-stimulated activation of its receptors with neutralizing VEGF antibodies (ranibizumab and bevacizumab) and the extracellular domains of VEGFR1 and 2 (aflibercept) is currently an important therapeutic approach to angiogenesis in these eye diseases but requires chronic treatment[8, 10]. Although these anti-VEGF agents can reduce neo-vascular growth and lessen vascular leakage, there are still therapeutic challenges to a significant number of patients with these eye diseases[11].

Adeno-associated viruses (AAVs) are small viruses that are not currently known to cause any disease, and their derived vectors show promise in human gene therapy[12, 13]. The clustered regularly interspersed palindromic repeats (CRISPR)-associated DNA endonuclease (Cas)9 in Streptococcus pyogenes (SpCas9) processes pre-crRNA transcribed from the repeat spacers into CRISPR RNAs (crRNA) and cleave invading nucleic acids on the guidance of crRNA and trans-activating crRNA (tracrRNA)[14, 15]. A single guide RNA (sgRNA) engineered as the crRNA-tracrRNA chimeric RNA can direct sequence-specific SpCas9 cleavage of double-strand DNA containing an adjacent "NGG" protospacer-adjacent motif (PAM)[14]. This CRISPR/Cas9 system is a powerful tool for the targeted introduction of mutations into eukaryotic genomes and subsequent protein depletion[16, 17].

In this study, we employed the AAV-mediated CRISPR/Cas9 system to edit genomic VEGFR2 in vivo and showed that editing of VEGFR2 abrogated angiogenesis in two mouse models of oxygen-induced retinopathy (OIR) and laser-induced choroid NV (CNV).

## Results

**CRISPR/Cas9-mediated depletion of VEGFR2 in vascular ECs in vitro**. Recombinant AAV (rAAV) vectors are at present the leading candidates for virus-based gene therapy thanks to their broad tissue tropism, non-pathogenic nature, and low immunogenicity[13]. In this study, we adapted a dual-AAV vector system packaging SpCas9 and SpGuide[16]. To identify an appropriate AAV serotype that could transduce vascular endothelial cells (ECs), we replaced the GFP promoter (phSyn) in the AAV-SpGuide vector[16] with a promoter of cytomegalovirus (CMV) (Fig. 1a)[15].

A major goal of gene therapy is the introduction of genes of interest into desired cell types. To circumvent targeting VEGFR2 in photoreceptors of eye tissues[18], an endothelial-specific promoter is designed to drive expression of SpCas9. Thus, we substituted the Mecp2 promoter in the AAV-pMecp2-SpCas9 vector[16] for an endothelial-specific promoter of intercellular adhesion molecule 2 (pICAM2)[19] (Fig. 1b).

Recombinant adeno-associated virus serotype 1 (rAAV1) has been shown to transduce vascular ECs in high efficiency[20]. We next examined whether rAAV1 was able to deliver the CRISPR–Cas9 into ECs[20, 21]. As shown in Fig. 1c, rAAV1 was able to infect human primary retinal microvascular ECs (HRECs), human primary umbilical vein ECs (HUVECs) as well as human primary retinal pigment epithelial cells (hPRPE). Subsequently, we transduced these cells with rAAV1-pICAM2-SpCas9 (rAAV1-SpCas9) for testing if the ICAM2 promoter was able to drive SpCas9 expression in ECs specifically. Western blot analysis of the transduced cell lysates indicated that SpCas9 was expressed in HRECs and HUVECs, but not in hPRPE cells (Fig. 1d), demonstrating that the dual vectors of AAV-SpCas9 and AAV-SpGuide are able to specifically target genomic loci of ECs. Then, a target mouse genomic sequence named as mK22 (Fig. 1a) corresponding to the most efficient sgRNA-targeting human VEGFR2 exon 3 named as K12 among the four target sequences[22] was cloned into the SpGuide vector.

To assess the editing efficiency of our dual-vector system in vitro, we infected C57BL/6 mouse primary brain microvascular ECs (MVECs) using rAAV1-SpCas9 with rAAV1-mK22 or rAAV1-lacZ. After 4 days post infection, the genomic DNA was isolated for PCR. Sanger DNA sequencing results showed that there were mutations around the PAM sequence of PCR products from MVECs transduced with rAAV1-SpCas9 plus -mK22 but not from those with rAAV1-SpCas9 plus -lacZ (Fig. 1e), suggesting that the mK22-guided SpCas9 cleaved the VEGFR2 locus at the expected site in MVECs. To find potential off-targets for the mK22-targeted genes, the "CRISPR Design Tool" (http://crispr.mit.edu/) was used. NGS analysis indicated that mK22 did not influence on the most possible off-target sequence in MVECs. Western blot analysis of the transduced cell lysates indicated that there was an 80% decrease in VEGFR2 from the transduced MVECs with SpCas9/mK22 compared with those with SpCas9/lacZ (Fig. 1f), demonstrating that the AAV-CRISRP/Cas9 system with mK22 efficiently and specifically induced mutations within the VEGFR2 locus and subsequent protein depletion in MVECs in vitro.

**Transduction of ECs with rAAV1 in vivo**. Gene delivery to the vasculature has significant potential as a therapeutic strategy for several cardiovascular disorders including atherosclerosis and angiogenesis. However, there is a pronounced challenge in achieving successful gene transfer in vascular ECs in vivo. To determine if rAAV1 was capable of transducing vascular ECs of NV in the C57BL/6 mouse models of OIR[23] and laser-induced CNV[24], we intravitreally injected rAAV1-CMV-GFP into mouse eyes at postnatal day 12 (P12) with or without experiencing the OIR model and immediately after the post-laser injury to Bruch's membranes of six-week-old mice in the CNV model, respectively. Whole-mount retinas of the P17 mice from the OIR model and the whole-mount choroids of the mice at day 7 after injection from the CNV model were stained with mouse endothelial-specific marker isolectin 4 (IB4)-Alexa 594. The merged images of IB4 with GFP indicated that rAAV1 was able to transduce normal vascular ECs in the retinal (Supplementary Fig. 1) and that preferentially transduced vascular ECs of NV induced by hypoxia and laser injury in the OIR (Fig. 2 and Supplementary Figs. 2 and 3) and CNV models (Fig. 2 and Supplementary Fig. 4), respectively.

**Editing genomic VEGFR2 abrogated hypoxia-induced angiogenesis**. To investigate whether the dual AAV system of AAV-SpCas9 and AAV-SpGuide (mK22) was able to edit VEGFR2 and inhibit pathological angiogenesis in vivo, we intravitreally injected equal amount of rAAV1-SpCas9 and rAAV1-mK22 or rAAV1-lacZ into P12 mouse eyes in the OIR mouse model[23]. In this model, P7 mouse pups with nursing mothers are subjected to hyperoxia (75% oxygen) for 5 days, which inhibits retinal vessel growth and causes significant vessel loss. On P12, mice are

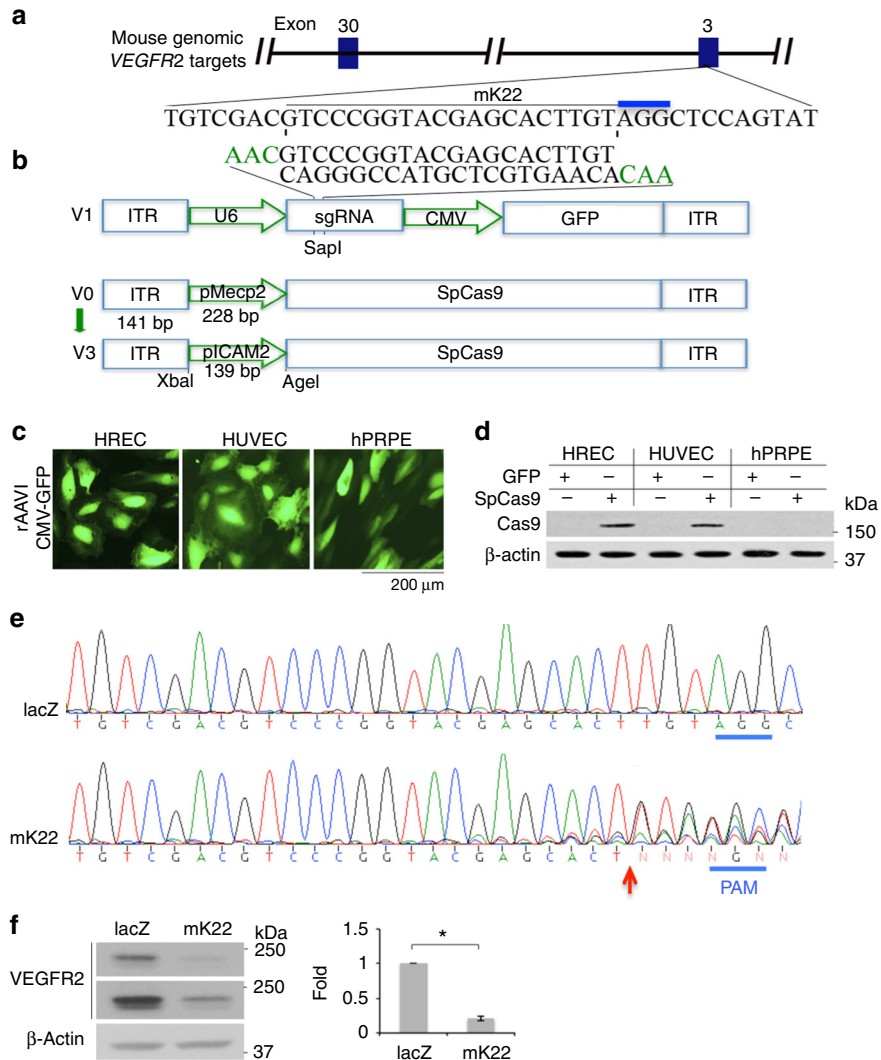

**Fig. 1** AAV-CRISPR/Cas9-mediated depletion of VEGFR2 in vitro. **a** Schematic of AAV-SpGuide (V1)[15]. Graphical representation of the mouse *VEGFR2*-targeted locus. The oligos of mK22 and its compliment were annealed and cloned into the V1 vector by SapI. The PAM is marked in *blue*. ITR inverted terminal repeat, U6 a promoter of polymerase III, CMV a promoter of cytomegalovirus, GFP green fluorescent protein. **b** Schematic of AAV-SpCas9 (V3). pMecp2: a neuron-specific promoter for methyl CpG-binding protein in V0 was substituted for pICAM2[19] by XbaI/AgeI. **c** Transduction of cultured cells with rAAV1. HRECs, HUVECs, and hPRPE cells in a 48-well plate to 50% confluence were infected with rAAV1-CMV-GFP (2 μl/well, 3.75 × 10[12] viral genome-containing particles (vg)/ml). Three days later, the cells were photographed under an immunofluorescence microscope. Three independent experiments showed rAAV1 transduction efficiency in HRECs, HUVECs and hPRPE cells of 85.6 ± 2.2, 88.5 ± 2.3 and 86.8 ± 2.6%, respectively. Scale bar: 200 μm. **d** pICAM2-driven expression of SpCas9 in ECs. After transduction with rAAV1-CMV-GFP (GFP) or rAAV1-pICAM2-SpCas9 (SpCas9) (2 μl/well, 3.75 × 10[12]vg/ml) in a 48-well plate for 4 days, cell lysates were subjected to western blot analysis with antibodies against Cas9 and β-actin. Data shown are representative of three independent experiments. **e** Sanger DNA sequencing was conducted on PCR products amplified from the genomic *VEGFR2* loci of MVECs, which were transduced by rAAV1-SpCas9 plus rAAV1-lacZ (lacZ) or rAAV1-mK22 (mK22). **f** Depletion of VEGFR2 expression using AAV-CRISPR/Cas9. Total cell lysates from the transduced MVECs were subjected to western blot analysis with antibodies against VEGFR2 and β-actin. The bar graphs are mean ± SD of three independent experiments. "*" indicates a significant difference between the compared two groups using an unpaired *t*-test. $p < 0.05$

returned to room air and the hypoxic avascular retina triggers both normal vessel regrowth and retinal NV named as preretinal tufts, which is maximal at P17[23]. Thus, on P17, the whole-mount retinas were stained with IB4. The results (Fig. 3a–c and Supplementary Fig. 5) showed that there was a dramatic decrease in the number of preretinal tufts and significantly more avascular areas from mice injected with rAAV1-SpCas9/mK22 than those with rAAV1-SpCas9/lacZ, suggesting that genome editing of *VEGFR2* by SpCas9/mK22 inhibits retinal NV in this OIR mouse model. Next-generation sequencing results (Fig. 3d) confirmed that there was about 2% insertion/deletions (indels) around the PAM from genomic DNA of the retinas treated with AAV-

SpCas9/mK22, but none with AAV-SpCas9/lacZ. In addition, western blot analysis of the retinal lysates showed that there was an about 30% reduction in VEGFR2 from mice treated with rAAV1-SpCas9/mK22 compared with controls (Fig. 3e, f). Taken together, these data demonstrate that editing genomic *VEGFR2* locus with SpCas9/mK22 abrogates hypoxia-induced angiogenesis in this OIR mouse model. In addition, the intravitreal injection of SpCas9/mK22 did not cause detectable damage to the retina morphology and function examined by optical coherence tomography (OCT), electroretinography (ERG), fluorescein fundus angiography (FFA), and whole-mounted retina staining by IB4 at the time point of 4 weeks (Supplementary Fig. 6).

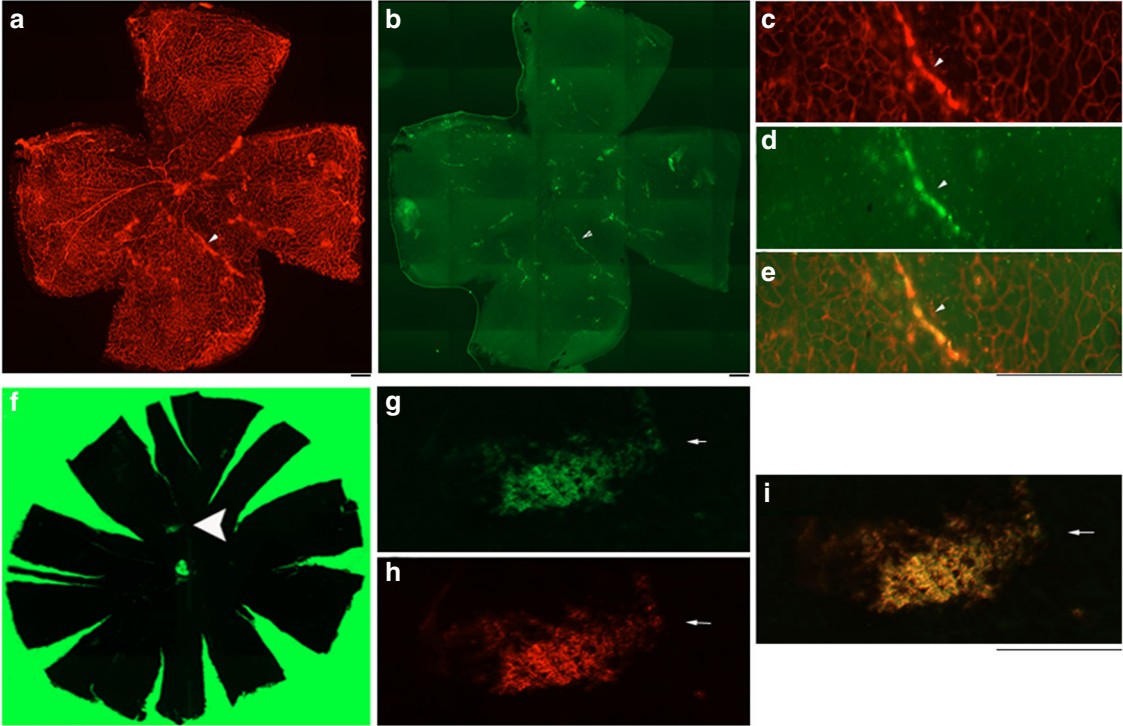

**Fig. 2** Transduction of ECs with rAAV1 in vivo. **a, b** On P7, C57BL/6J litters were exposed to 75% oxygen until P12[23, 37] when the pups were injected intravitreally with rAAV1-CMV-GFP (1 μl, 3.75 × 10[12] vg/ml). After return to room air (21% oxygen) for 5 days, and whole-mount retinas from the killed mice were stained with IB4 (*red*). Images were taken under TxRed channel (**a, c**), GFP channel (**b, d**). **e** Merged image of **c** and **d**. Scale bar: 200 μm. **f–i** Four lesions were induced in an eight-week-old mouse on the Bruch's membrane using a 532-nm green laser. rAAV1-CMV-GFP (1 μl, 3.75 × 10[12] vg/ml) was injected intravitreally into the mouse. Seven days later, the whole-mount choroid was stained with IB4, and images were taken under GFP channel (**f, g**) and Txred channel (**h**). **i** A merged image of **g** and **h**. Scale bar: 200 μm. Each figure represents at least six ones from different mice

**VEGFR2 suppressed NV in laser-induced CNV in mice.** We also assessed whether the rAAV1-SpCas9/mK22 could inhibit NV in the laser-injury-induced CNV mouse model, which has been used extensively in studies of the exudative form of human AMD[24]. First, we intravitreally injected rAAV1-SpCas9 with rAAV1-mK22 or rAAV1-lacZ into mouse eyes following the laser injury. In this model, NV grows from choroid vessels after laser injury on Bruch's membrane, and on day 7 there is the maximal CNV, which begins to regress spontaneously after 14–21 days[24]. Hence, on day 7, fluorescein was injected into the mice intraperitoneally, and images of fluorescein angiography (FA) were taken. Subsequently, the flat-mount choroids were stained by IB4 for analysis of laser-injury-induced CNV. As shown in Fig. 4a–c, there was less NV in the eyes injected with rAAV1-SpCas9/mK22 than those with rAAV1-SpCas9/lacZ on day 7.

To examine if editing genomic VEGFR2 could promote regression of CNV, rAAV1s were intravitreally injected on day 7 in the mouse CNV. On day 14, the images of FA and IB4 staining showed that there was less CNV from the mice injected with rAAV1-SpCas9/mK22 than those with rAAV1-SpCas9/lacZ (Fig. 4d–f). These data indicate that editing the genomic VEGFR2 locus with SpCas9/mK22 suppresses NV in this laser-injury-induced CNV model. Taken together, our data establish a strong foundation for genome editing as a novel therapeutic approach to angiogenesis-associated diseases.

## Discussion

We report that rAAV1 preferentially transduced vascular ECs of pathological vessels in both mouse models of OIR and laser-injury induced CNV (Fig. 2 and Supplementary Fig. 3) while also transducing normal vascular ECs in the retina (Supplementary

Fig. 1). The preferential transduction of ECs in pathological vessels may be due to the fact the neovessels are less mature than normal vessels, and have incomplete basement membrane and weaker intercellular junctions. To date, AAV vectors have been used in a number of clinical trials such as for Leber's congenital amaurosis[25–27] and congestive heart failure[28] and has been approved for treatment of lipoprotein lipase deficiency in Europe[29, 30]. While anti-VEGF agents (e.g., ranibizumab and aflibercept) can reduce NV growth and vascular leakage-associated eye diseases (e.g., PDR and wet AMD), therapeutic challenges remain, including the need for chronic treatment and a significant number of patients who do not respond[11]; gene therapy targeting genomic VEGFR2 using AAV-CRIPSR/Cas9 may provide a novel alternative approach. While other genes, such as MMP9[31, 32], have been linked to various proliferative retinopathies, none has been shown to drive new vessel disease to the extent seen VEGFR2.

Success translation of genome editing technologies to the clinic must address some major obstacles, primarily in terms of the safety and efficacy; genetic modifications are permanent, and deleterious off-target mutations could create cells with oncogenic potential, reduced cellular integrity, and or functional impairment[33, 34]. Our results demonstrate that expression of VEGFR2 was depleted by 80% in vitro (MVECs) (Fig. 1) and by 30% in vivo (retina) (Fig. 3) by the AAV-CRISRP/Cas9 (mK22), in which SpCas9 was driven by an endothelial cell-specific promoter pICAM2 (Fig. 1). In addition, NGS analysis indicated that there was only about 2% indels around the PAM in the PCR products amplified from the treated P17 mouse retinas, and there was a significant decrease in NV in both mouse models of OIR (Fig. 3) and CNV (Fig. 4) after treatment with AAV-CRISPR–Cas9-targeting genomic VEGFR2 in comparison to targeting control lacZ.

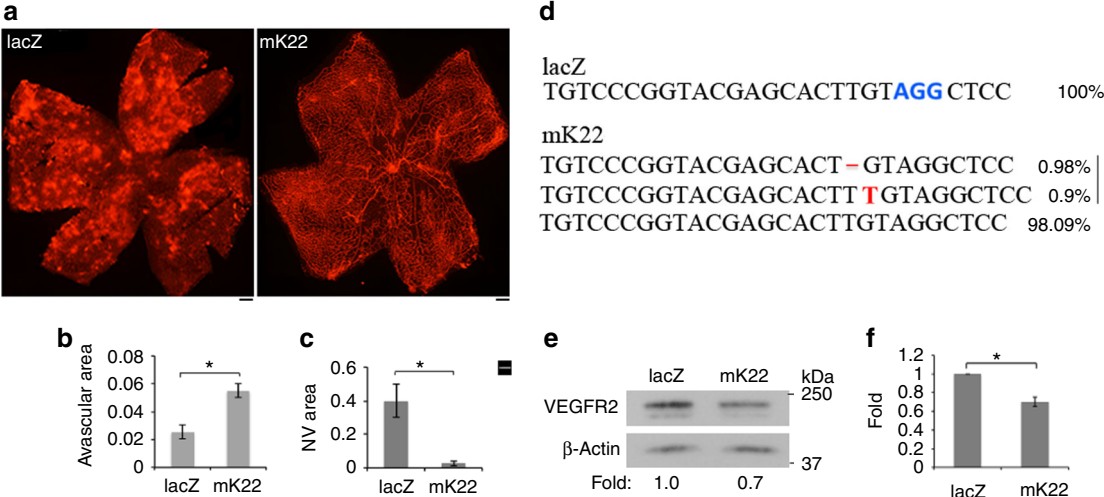

**Fig. 3** Editing genomic VEGFR2 abrogated hypoxia-induced angiogenesis. **a** Litters of P12 mice that had been exposed to 75% oxygen for 5 days were injected intravitreally with 1 μl ($3.75 \times 10^{12}$ vg/ml) containing equal rAAV1-SpCas9 and rAAV1-lacZ (lacZ) or rAAV1-mK22 (mK22). On P17, whole-mount retinas were stained with IB4. lacZ and mK22 indicate retinas from the rAAV1-SpCas9/lacZ and mK22-injected mice, respectively. **b** Analysis of avascular areas from the IB4-stained retinas ($n = 6$). **c** Analysis of NV areas from the IB4-stained retinas ($n = 6$). **d** NGS analysis of indels. The DNA fragments around the PAM sequences were PCR amplified from genomic DNA of the rAAV1-SpCas9/lacZ or -mK22-injected retinas, and then subjected to NGS. **e, f** The lysates of the rAAV1-SpCas9/lacZ or -mK22-injected retinas were subjected to western blot analysis using indicated antibodies. The bar graph data are mean ± SD of three retinas. "*" indicates significant difference using an unpaired t-test. $p < 0.05$

In summary, our studies show that precise and efficient gene editing of *VEGFR2* using CRISPR–Cas9 systems has the potential to treat angiogenesis-associated diseases.

## Methods

**Mice.** Six–eight-week-old mice (C57BL/6J, male and female) were purchased from Jackson Laboratories (Bar Harbor, ME). All the animal experiments followed the guidelines of the Association for Research in Vision and Ophthalmology Statement for the Use of Animals in Ophthalmic and Vision Research. There were at least three experiments for statistic analyses and investigators who conducted analysis were masked as to the treatment groups. All the mice were cared for by following the ACUC protocol approved by the Institutional Animal Care and Use Committee at Schepens Eye Research Institute.

**Major reagents.** Antibodies against VEGFR2 (1:1000 for western blot) and β-actin (1:5000 for western blot) were purchased from Cell Signaling Technology (Danvers, MA) and Santa Cruz Biotechnology (Santa Cruz, CA), respectively. Horseradish peroxidase (HRP)-conjugated goat anti-rabbit IgG (1:5000) and anti-mouse IgG (1:5000) were from Santa Cruz Biotechnology (Santa Cruz, CA). Enhanced chemiluminescent substrate for detection of HRP was from ThermoFisher Scientific (Waltham, MA). Alexa fluorescence- 594-conjugated mouse endothelial-specific isolectin B4 (IB4) was purchased from Life Technology (Grand Island, NY). The plasmids of pAAV-pMecp2-SpCas9-spA (AAV-SpCas9) (cat. no. 60,957) and pAAV-U6-sgRNA(SapI)-hSyn-GFP-KASH-bGH (SpGuide acceptor) (cat. no. 60,958) were purchased from Addgene (Cambridge, MA). High-fidelity Herculase II DNA polymerases were from Agilent Technologies (Santa Clara, CA).

**Cell culture.** C57BL/6 mouse primary brain microvascular endothelial cells (MVECs) were purchased from CellBiologics (cat. no. C57-C57-6023, Chicago, IL) and cultured in the endothelial cell medium with a kit (CellBiologics). Human primary retinal microvascular endothelial cells (HRECs) were purchased from Cell Systems (cat. no. ACBR1 181V, Kirkland, WA) and cultured in endothelial growth medium-2 (Lonza, Walkersville, MD). Primary human umbilical vein endothelial cells (HUVECs, cat. no. CC-2517) and human primary retinal pigment epithelial cells (hPRPE, cat. no. 194987) were purchased from Lonza. HUVECs were cultured in Medium 199 (Sigma) supplemented with 20% bovine calf serum (HyClone, Logan, UT), 100 mg/ml heparin, 12 mg/ml bovine brain extract (Hammond Cell Tech, Windsor, CA). hPRPE cells were cultured in a 1:1 mixture of low-glucose Dulbecco's modified Eagle's medium (Life Technologies, Grand Island, NY) and Ham's F-12 Nutrient Mixture (Gibco) supplemented with 10% fetal bovine serum (Lonza, Walkersville, MD). Tissue culture dishes were pre-coated with 0.2% gelatin in phosphate-buffered saline for MVECs, HRECs, and HUVECs[35]. All cells were cultured at 37 °C in a humidified 5% $CO_2$ atmosphere[36].

**DNA constructs.** The 20 nt target DNA sequence (5′-GTCCCGGTACGAG-CACTTGT-3′, mK22) preceding a 5′-NGG PAM sequence at exon 3 in the mouse *VEGFR2* genomic locus (NC_000071.6) was selected for generating single guide RNA (sgRNA) for SpCas9 using the CRISPR design tool. The control sgRNA sequence (5′-TGCGAATACGCCCACGCGATGGG-3′) was designed to target the *lacZ* gene of *Escherichia coli*[16]. The pAAV-U6-sgRNA-CMV-GFP vector (V1) was originated from AAV-SpCas9 (cat. no. 60,958)[16] by replacing the hSyn-GFP with the PCR-amplified CMV-GFP from pEGFP-C1 vector (Clontech, cat. no. 6084-1) using XbaI/EcoRI as described previously[15]. The pAAV-pICAM2-SpCas9 (V3) was derived from AAV-SpGuide (cat. no. 60,957) by replacement of the promoter pMecp2 using XbaI/AgeI with pICAM2, which was PCR amplified from genomic DNA isolated from HRECs. The PCR primers for this amplification were: forward 5′-CGTCTAGAGTAGAACGAGCTGGTGCACGTGGC-3′, reverse 5′-GGACCGGTCCAAGGGCTGCCTGGAGGGAG-3′. All these constructs were confirmed by DNA sequencing.

To construct SpGuides, the top oligo 5′-ACC-GTCCCGGTACGAGCACTTGT-3′) and bottom oligo: 5′-AAC-20nt-C-3′ (20 nt: complimentary target *mK22* DNA sequences) were annealed and cloned into the V3 vector by *SapI*. All clones were confirmed by DNA sequencing using a primer 5′-GGACTATCATATGCTTACCG-3′ from the sequence of U6 promoter, which drives expression of sgRNAs.

Both synthesis of primers and oligos and sequencing of PCR products and clones were done by Massachusetts General Hospital (MGH) DNA Core Facility (Cambridge, MA).

**Production of AAVs.** The recombinant AAV2/1 (rAAV1) vectors were produced as described previously[17] in the Gene Transfer Vector Core in Schepens Eye Research Institute of Massachusetts Eye and Ear (Boston, MA). Briefly, triple transfection of AAV package plasmid (AAV2/1), transgene plasmid (pAAV-pICAM2-SpCas9: AAV-SpCas9, pAAV-U6-mK22-CMV-GFP: AAV-mK22 or pAAV-U6-*lacZ*-CMV-GFP: AAV-lacZ) and adenovirus helper plasmid were performed in a 10-layer hyper flask containing confluent HEK 293 cells. At day 3 post transfection, the cells and culture medium were collected and enzymatically treated with Benzonase (EMD Millipore). After high-speed centrifugation and filtration, the cell debris was cleared. The viral solution was concentrated by running through tangential flow filtration, and then loaded onto an iodixional gradient column. After one round of ultracentrifugation, the pure vectors were separated and extracted, then ran through an Amicon Ultra-Centrifugal Filter device (EMD Millipore) for desalting. Both vectors were titrated by TaqMan PCR amplification (Applied Biosystems 7500, Life Technologies), with the primers and probes detecting the transgene. Sodium dodecyl sulfate-polyacrylamid gel electrophoresis (SDS-PAGE) was performed to check the purity of the vectors, which were named rAAV1-SpCas9, rAAV1-mK22, and rAAV1-lacZ.

**Transduction of cultured cells.** MVECs, HRECs, HUVECs, and hPRPE cells grown to 50% confluence in a 48-well plate were changed into the fresh cultured media and added either with rAAV1-mK22, rAAV1-lacZ, rAAV1-SpCas9

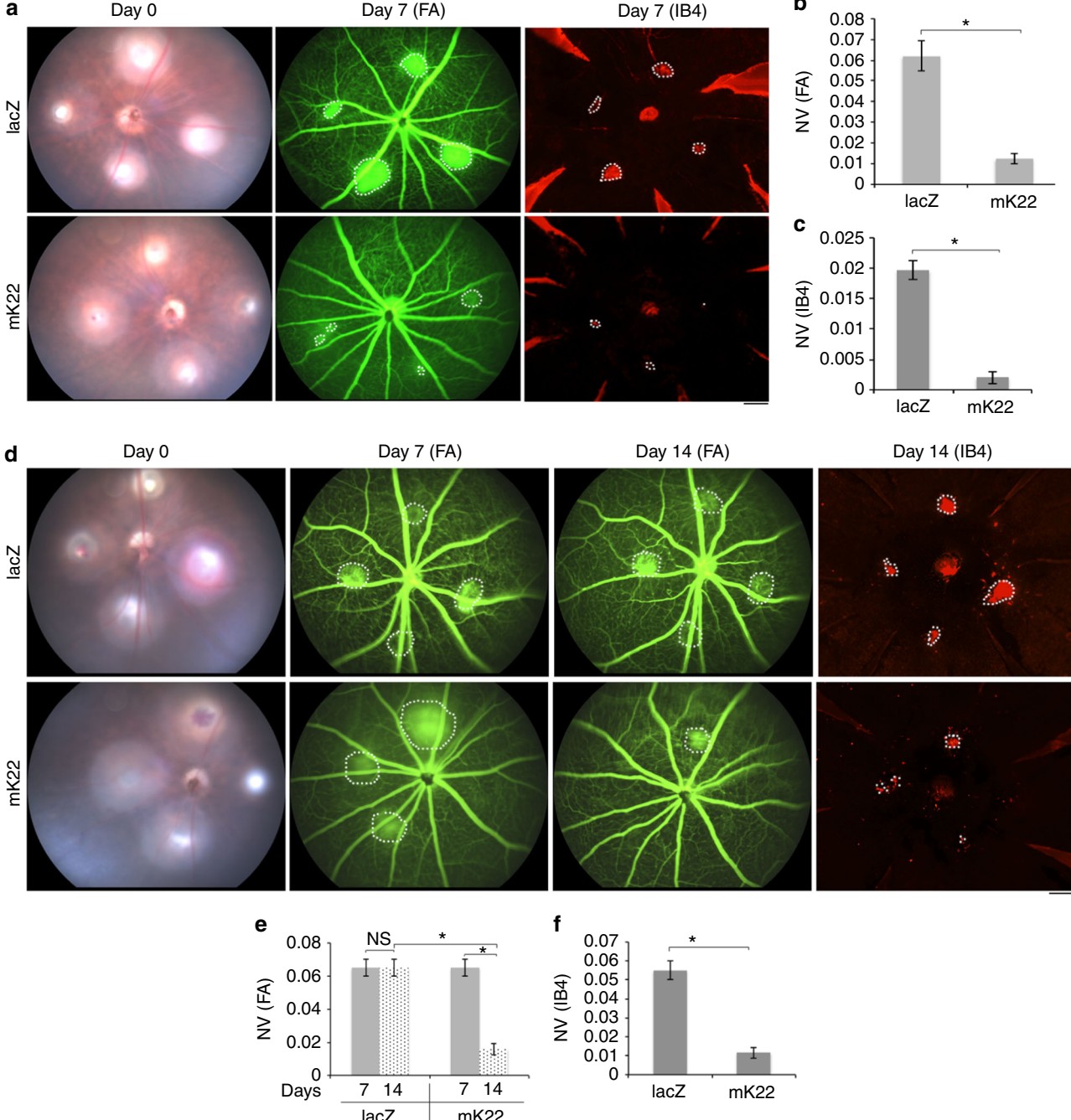

**Fig. 4** AAV-CRISPR/Cas9-targeting genomic *VEGFR2* suppressed NV in laser-induced choroid NV in mice. After laser injury of Bruch's membrane, fundus images (day 0) were taken using the Micron III system, and the mice were injected intravitreally with 1 μl (3.75 × 10^12 vg/ml) containing equal rAAV1-SpCas9 and rAAV1-lacZ or -mK22 right immediately after the laser injury (**a**) or 7 days of the laser injury (**d**). Seven days after AAV1 injection, the mice were injected intraperitoneally with fluorescein, and the FA images were taken using the Micron III system. Subsequently, whole mounts of choroids were stained with IB4, and the images were taken under an immunofluorescence microscope. Areas of NV were analyzed based on the images of FA (**b**, **e**) and IB staining (**c**, **f**) (*n* = 6). "*" indicates significant difference between the compare two groups using an unpaired *t*-test. *p* < 0.05

individually or both of rAAV1-SpCas9 with rAAV1-mK22 or rAAV1-lacZ (2 μl/well for each rAAV1, 3.75 × 10^12 viral genome-containing particles (vg)/ml). Three days later, the cells were photographed under an immunofluorescence microscope for determining the rAAV1 transduction efficiency. After 4 days, the cells were lysed with 1× sample buffer for western blotting analysis or collected for genomic DNA isolation.

**Western blot**. Cells were lysed in 1× sample buffer, which was diluted with extraction buffer (10 mM Tris-HCl, pH 7.4, 5 mM EDTA, 50 mM NaCl, 50 mM NaF, 1% Triton X-100, 20 μg/ml aprotinin, 2 mM Na$_3$VO$_4$, and 1 mM phenylmethylsulfonyl fluoride) from the 5× protein sample buffer (25 mM EDTA (pH

= 7.0), 10% sodium dodecyl sulfate (SDS), 500 mM dithiothreitol, 50% sucrose, 500 mM Tris-HCl (pH = 6.8), and 0.5% bromophenol blue). The lysates were boiled for 5 min and then centrifuged for 5 min at 13,000 × *g*. Proteins from the samples were separated by 10% SDS-PAGE, transferred to polyvinylidene difluoride membranes, and subjected to western blot analysis. Experiments were repeated at least three times. Signal intensity was determined by densitometry using NIH ImageJ software[36].

**DNA sequencing**. Cells were collected for genomic DNA extraction using the QuickExtract DNA Extraction Solution (Epicenter, Chicago, IL), following the manufacturer's protocol. In brief, the pelleted cells were re-suspended in the

QuickExtract solution, vortexed for 15 s, incubated at 65 °C for 6 min, vortexed for 15 s and then incubated at 98 °C for 10 min. The genomic region around the PAM was PCR amplified with high-fidelity Herculase II DNA polymerases. The PCR primers were (forward 5′-GCTCCTGTCGGGTCCCAAGG-3′) and (reverse 5′-ACCTGGACTGGCTTTGGCCC-3′). The PCR products were separated in 2% agarose gel and purified with a gel extraction kit (Thermo Scientific) for Sanger DNA sequencing and NGS[15]. DNA sequencing was performed by the MGH DNA core facility.

**A mouse model of OIR**. C57BL/6J litters on postnatal day (P)7 were exposed to 75% oxygen until P12 in the oxygen chamber (Biospherix). Oxygen concentration was monitored daily using an oxygen sensor (Advanced Instruments, GPR-20F)[23, 37]. On P12, the pups were anesthetized by intraperitoneal injection of 50 mg/kg ketamine hydrochloride and 10 mg/kg xylazine. During intravitreal injections, eyelids of P12 pups were separated by incision. Pupils were dilated using a drop of 1% tropicamide and the eyes were treated with topical proparacaine anesthesia. Intravitreous injections were performed under a microsurgical microscope using glass pipettes with a diameter of ~150 μm at the tip after the eye were punctured at the upper nasal limbus using a a BD insulin syringe with the BD ultra-fine needle. One μl of rAAV1-CMV-GFP or both of rAAV1-SpCas9 with rAAV1-mK22 or rAAV1-lacZ (1 μl, 3.75 × 10^12 vg/ml) was injected. After the intravitreal injection, the eyes were treated with a triple antibiotic (Neo/Poly/Bac) ointment and kept in room air (21% oxygen). On P17, the mice were killed and retinas were carefully removed and fixed in 3.7% paraformaldehyde (PFA), and the mice under 6 g were excluded from the experiments. In total, there were three experiments performed in this OIR model. Retinal whole mounts were stained overnight at 4 °C with murine-specific EC marker isolectin 4 (IB4)-Alexa 594 (red)[23, 38, 39]. The images were taken with an EVOS FL Auto microscope (Life Technologies).

**Quantification of vaso-obliteration and NV**. This was performed previously[23]. Briefly, retinal image was imported into Adobe Photoshop CS4, and the Polygonal Lasso tool was used to trace the vascular area of the entire retina. Once the vascular area was highlighted, the number of pixels was obtained. After selecting total retinal area, the Lasso tool and the "subtract from selection" icon was used to selectively remove the vascularized retina, leaving behind only the avascular area. Once the avascular region was selected, click the refresh icon again to obtain the number of pixels in the avascular area.

When analyzing NV, the original image was reopened. The magic wand tool was selected from the side tool panel on the left side of the screen. On the top tool panel, the tolerance to a level that will pick up NV was set while excluding normal vessels (beginning at 50). Regions of NV were selected by clicking on them with the magic wand tool. The areas of NV fluoresced more intensely than surrounding normal vessels. When neovessels were selected, the area of interest was zoomed in by holding the "Alt" key on the keyboard and scrolling up. When all NV was selected and checked, the refresh icon recorded the total number of pixels clicked in the NV area.

**Laser-induced choroid NV in mice**. Ten mice (Stock number: 664, C57BL/6J, male and female, 17–22 g, 6–8 weeks old, Jackson Laboratories, Bar Harbor, ME) were deeply anesthetized with an intraperitoneal injection of ketamine/xylazine (120 mg/kg ketamine/20 mg/kg xylazine). Their pupils were be dilated using a drop of 1% tropicamide and the eye were treated with topical proparacaine anesthesia drops. The mice were placed on a specialized stage with the Micron III retina imaging system (Phoenix Research Labs, Pleasanton, CA) using Genteal gel (Novartis, Basel, Switzerland). Under real-time observation, laser photocoagulation were applied to the eyes using a Streampix5 laser system (Meridian AG, Zürich, Switzerland) at 532 nm wavelength (100 μm of diameter, 0.1 s of duration and 100 mW of power). Four lesions located at the 3, 6, 9, and 12 o'clock meridians around the optic nerve were induced. Laser-induced disruption of Bruch's membrane was identified by the appearance of a bubble at the site of photocoagulation. Fundus images were taken on the anesthetized mice using the Micron III retina imaging system with illumina light. Laser spots that did not result in the formation of a bubble were excluded from the studies. Laser spots were also be confirmed by OCT[24, 40]. rAAV1 (1 μl, 3.75 × 10^12 vg/ml) was injected into the vitreous using glass pipettes with fine tips after puncturing the sclera 1 mm from the limbus with a 30-gauge needle under an operation surgical microscope. On day 7 or 14, animals were anesthetized as described above. Fundus images were taken using the Micron III retina imaging system with illumina light. Then 0.01 ml of 25% sodium fluorescein (pharmaceutical grade sodium fluorescein; Akorn Inc) per 5 g body weight was injected intraperitoneally. The retinal vasculature filled with dye in <1 min following injection. Images of FA were taken with UV light sequentially at 2 and 5 min post-fluorescein injection. Seven days after rAAV1 injection, the mice were killed, and the mouse eyes were carefully removed and fixed in 3.7% PFA. Whole-mount choroids were stained overnight at 4 °C with IB4[23, 38, 39]. The images were taken with an EVOS FL auto microscope.

**NGS analysis of potential off-targets**. To find potential off-targets for the mK22-targeted genes, the "CRISPR Design Tool" (http://crispr.mit.edu/) was used[16], indicating that the most potential off-target sequence was 5′-

CTCACGGTTGGAGCACTTGTAGG-3′ that was located at Chr7:-126856352. On the basis of this information, we designed PCR primers (forward primer P25F: 5′-AGCTTCATTCAGTGTCTCTGGG-3′, reverse primer P25R: 5′-GGGTATTTG-TAAGGTGCTGTTGA-3′) for PCR amplification of the DNA fragment covering the potential mK22 off targets. The PCR products from MVECs transduced by the dual AAV-CRISPR/Cas9 vectors either containing lacZ-sgRNA or VEGFR2-sgRNA (mK22) were sent for Sanger DNA sequencing and NGS.

**Examination of toxicity of the dual AAV-CRISPR/Cas9 in mouse eyes**. On P12, five pups were anesthetized and underwent intravitreal injections as described above. During injection, One μl of rAAV1-SpCas9 plus rAAV1-mK22 was injected.

After 4 weeks, OCT was performed using a spectral domain (SD-) OCT system (Bioptigen Inc., Durham, NC). Briefly, mice were deeply anesthetized with an intraperitoneal injection of ketamine/xylazine (100–200 mg/kg ketamine/20 mg/kg xylazine). The pupils were dilated with topical 1% tropicamide to view the fundus. After anesthesia, Genteal gel was applied to both eyes to prevent drying of the cornea. The fundus camera in the optical head of the apparatus provided initial alignment for the sample light, to ensure it is delivered through the dilated pupil. Final alignment was guided by monitoring and optimizing the real-time OCT image of the retina, with the whole set-up procedure taking ~5 min for each mouse eye.

At week 4, after OCT, ERG (by light/dark adaptation, using a DIAGNOSYS ColorDome containing an interior stimulator) was performed as followed. Following overnight dark adaptation, the animals were prepared for ERG recording under dim red light. While under anesthesia with a mixture of ketamine (100–200 mg/kg i.p.) and xylazine (20 mg/kg i.p.), their pupils were dilated using a drop of 1% tropicamide followed by a drop of 1% cyclopentolate hydrochloride applied on the corneal surface. One drop of Genteal (corneal lubricant) was applied to the cornea of the untreated eye to prevent dehydratation. A drop of 0.9% sterile saline was applied on the cornea of the treated eye to prevent dehydration and to allow electrical contact with the recording electrode (gold wire loop). A 25-gauge platinum needle, inserted subcutaneously in the forehead, served as reference electrode, while a needle inserted subcutaneously near the tail served as the ground electrode. A series of flash intensities was produced by a Ganzfeld controled by the Diagnosys Espion 3 to test both scotopic and photopic response.

The following day after ERG, FFA was performed on the mice. Animals were anesthetized with a mixture of ketamine (100–200 mg/kg i.p.) and xylazine (20 mg/kg i.p.), and their pupils were dilated using a drop of 1% tropicamide and the eye will be treated with topic anesthesia (Proparacaine drops). A drop of sterile saline was placed on the experimental eye to remove any debris followed by Genteal. Genteal was placed on both eyes to prevent corneal drying. Then 0.01 ml of 25% sodium fluorescein (pharmaceutical grade sodium fluorescein; Akorn Inc) 5 g body weight was injected i.p. The retinal vasculature was filled with dye in <1 min following injection. Photos were taken sequentially at 1, 2, 3, 4, and 5 min post-fluorescein injection. A Micron III (Phoenix Research) system was used for taking fundus photographs according to the manufacturer's instructions. The mice were placed in front of the Fundus camera and pictures of the retina taken for monitoring retinal function.

After the were killed, retinas were carefully removed and fixed in 3.7% PFA. Retinal whole mounts were stained overnight at 4 °C with murine-specific EC marker isolectin 4 (IB4)-Alexa 594 (red)[23, 38, 39]. The images were taken with an EVOS FL Auto microscope (Life Technologies).

**Statistics**. The data from three independent experiments in which the variance was similar between the groups were analyzed using an unpaired and two tailored t-test. For animal experiments, at least the data from six mice were used for the statistic analysis. p-values of <0.05 were considered statistically significant. All relevant data are available from the authors.

**Data availability**. The data supporting the findings of this study are available from the corresponding author on reasonable request.

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

## Acknowledgements

This work was supported by National Institutes of Health, National Eye Institute Grants R01 EY012509 (to H.L.) and Core Grant P30EY003790.

## Author contributions

X.H., G.Z., W.W., Y.D., G.M. and J.S. performed most of the experiments and analyzed the results. R.X. and L.V. contributed to production of AAVs. F.Z. contributed to experimental design and P.D.A. revised the manuscript. H.L. designed and performed experiments, analyzed the results, and wrote the manuscript.

## Additional information

**Competing interests:** The authors declare no competing financial interests.



