## [Peer Review File · Nature Communications]

Reviewers' comments:

Reviewer #1 (expert in gene editing)

Remarks to the Author:

In this manuscript the authors have performed SpCas9-based genome editing to deplete VEGFR2 in endothelial cells in vivo. An ICAM2 promoter driven Cas9 in rAAV1 was developed to target endothelial cells. The authors demonstrated only 2% indels in P17 mouse retinas and ~30% protein depletion. Significant change in avascular areas was reported. This study lacks significant novelty and technical advantage compared to published AAV-CRISPR studies. As described below, there are several major experiments and analyses that should be performed to support the main conclusions.

Major points:

- In Fig. 3D, the 2% indel rate is very low. NGS analysis indicated that there was only about 2% indels in treated P17 mouse retinas but protein of VEGFR2 was depleted by 30%. Since only a low percentage of the Vegfr2 genes are knocked out in the AAV infected retina, why is there a much higher protein depletion and retina pathology?
- In Fig. 2A-B, the authors did not quantify the ratio of endothelial cells that were infected by AAV (GFP+). It appears that GFP+ rate was low. Is there a delivery barrier to infect ECs in vivo using AAV? The authors should improve the infection efficiency.
- Only one Vegfr2 sgRNA was used. The authors did not measure any off-target effects in this study. Two sgRNAs should be used to rule out off-target effect.

Reviewer #2 (expert in retinal angiogenesis and associated diseases)

Remarks to the Author:

Huang et al present an interesting study on controlling angiogenesis using a CRISPR/Cas9 editing of VEGFR2. They evaluated the strategy in an oxygen-induced retinopathy model (OIR) as well as a laser-induced choroidal neovascularization model (CNV). The technology is state of the art but as written it is not clear what sequence is edited. That should be clarified at the end of Paragraph 4 of the Results section. The results are impressive and suggest that this is a viable strategy for controlling angiogenesis. The only major issue I have with the study is the use of the rAAV1-CMV-GFP control while rAAV1-ICAM2-SpCas9 is the therapeutic vector. CMV promoters are notorious for a short half life. The control should be rAAV1-ICAM2-GFP.

Other comments:

Figure 2. It appears that transfection occurs in large blood vessels not capillaries. Optimally, the therapy should target capillaries not large blood vessels. Are capillaries transfected and the photo is not good enough to see that or is there truly no transfection of capillaries?

Reviewer #3 (expert in AAV-based gene therapy)

Remarks to the Author:

The authors deliver the SpCas9 system in dual AAVs in order to test for depletion of VEGFR2 in vascular endothelial cells (ECs). They drove expression of SpCas9 with an endothelial-specific promoter (intercellular adhesion molecule 2 (ICAM2)) and use AAV serotype 1 (AAV1) to target endothelial cells.

They show that they can target human ECs in vitro and ECs of pathologic vessels in mouse models. They tested for efficacy (diminution of pathologic angiogenesis) in two mouse models of retinal neovascularization: the retinopathy of prematurity (ROP) mouse and the laser photocoagulation model. The AAVs were delivered intravitreally 5-7 days before the predicted time of maximal neovascularization. They found that the experimental AAVs (but not LacZ control) significantly diminished aberrant blood vessel formation.

The results are interesting and exciting but because this is one of the first papers using gene editing to manipulate an ocular disease model, it would be helpful to have some additional information:

1) There is no longterm follow-up data, presumably because the animal models are acute ones. But it would be helpful to know whether permanent expression of the transgenes resulted in any toxicity, particularly as they probably have not reached peak expression by the time that the readouts were made (5-7 days after injection). In particular, does production of the SpCas9 cause any toxicity/immune response?

2) Does the system interfere with normal blood vessel growth? I presume that the timepoint for delivery in the ROP model (P12) was selected to minimize this possibility. Should this be discussed?

3) Interestingly, in the laser photocoagulation model, intravitreal injection of the reagents ameliorates the outer retinal phenotype. Is there evidence that AAV1 penetrates across the mouse retina from the vitreal aspect or could it penetrate through the laser-damaged regions of the retina?

4) How could this approach be extrapolated to treat humans? For choroidal neovascularization, the reagents would presumably have to be injected into the subretinal space in direct contact with the aberrant blood vessels. Would the reagents be delivered after the disease process has initiated or before (as with ROP mouse)? If the reagents target mainly pathologic vessels, that may be the case. Also, do you expect that the sequence used to target the mouse VEGFR2 sequence would be the same target used for human disease (ie, are they homologous)? What steps would be necessary to test a sequence that could be applied to humans?

5) Relevant to the discussion about need for additional therapies, for humans who do not respond to the available therapies for ocular neovascularization, is there any data suggesting that there are other pathways/targets (besides VEGF) that could be awry?

Other:

1) Results: "Transduction of ECs....", ~3 sentences from end of section: brush' membrane? I assume this is Bruch's.

Reviewer #1

The authors demonstrated only 2% indels in P17 mouse retinas and ~30% protein depletion. Significant change in avascular areas was reported. This study lacks significant novelty and technical advantage compared to published AAV-CRISPR studies. As described below, there are several major experiments and analyses that should be performed to support the main conclusions.

Our report is one of the first papers using gene editing to manipulate an ocular disease model as the reviewer #3 points out; in addition, we used an ICAM2 promoter in the AAV vector to drive expression of SpCas9 specifically in endothelial cells. These features make this study innovative and creative.

Major points:

- In Fig.3D, the 2% indel rate is very low. NGS analysis indicated that there was only about 2% indels in treated P17 mouse retinas but protein of VEGFR2 was depleted by 30%. Since only a low percentage of the Vegfr2 genes are knockedout in the AAV infected retina, why is there a much higher protein depletion and retina pathology?

The retina contains a number of cell types including photoreceptors, ganglion cells, Müller cells, pericytes, and ECs, but expression of SpCas9 was driven by an endothelial cell specific promoter (pICAM2). Thus, only a very small population of cells in the retina were targeted by the dual AAV system. Among the cells in the retina, many cells do not express VEGFR2, while cells (e.g. photoreceptors) so express VEGFR2 but are not targeted by the CRISPR/Cas9. In addition, rAAV1 preferentially target the ECs in the pathological vessels. Thus, the results (2% of indels, 30% of VEGFR2 depletion, and inhibition of retina pathology) are plausible.

- In Fig. 2A-B, the authors did not quantify the ratio of endothelial cells that were infected by AAV (GFP+). It appears that GFP+ rate was low. Is there a delivery barrier to infect ECs in vivo using AAV? The authors should improve the infection efficiency.

We have quantified the proportion of GFP positive ECs (IB4 positive ECs). The results showed that GFP+ rate indeed was low ($4.3\% \pm 0.006$) (supplemental figure 3). In addition, we also quantified the GFP positive ECs vs the ECs in the pathological vessels. The results showed that rAAV1 infected almost all the ECs in the pathological vessels in the mouse models of OIR.

As shown in supplemental figure 1, rAAV1 infects ECs in the normal retina in vivo, suggesting there is not a delivery barrier to infect ECs in vivo using rAAV1.

- Only one Vegfr2 sgRNA was used. The authors did not measure any off-target effects in this study. Two sgRNAs should be used to rule out off-target effect.

At the beginning of this project, we selected four sgRNAs in exon 3 and four sgRNAs in exon 15 for targeting Vegfr2, and found this sgRNA worked most efficiently for depletion of VEGFR2 among the eight sgRNAs. As suggested, we measured off-target effects of this sgRNA. The results showed that there were no off-target effects detected by next generation sequencing of the most possible off-target, which was selected using the online tool crispr.mit.edu. We have included these data in this revision.

Reviewer #2

The technology is state of the art but as written it is not clear what sequence is edited. That should be clarified at the end of Paragraph 4 of the Results section.

We have clarified this point at the end of paragraph 4 of the Results. In the mouse VEGFR2 of NC_000071.6, next generation sequencing indicated that there was a T deletion in the position of 37884 (0.98%) and a T insertion (0.90%) between the nucleotides 37883 and 37884. Both the deletion and insertion occurred at the third position prior to the PAM, which is located in exon 3 of genomic VEGFR2.

The results are impressive and suggest that this is a viable strategy for controlling angiogenesis. The only major issue I have with the study is the use of the a rAAV1-CMV-GFP control while rAAV1-ICAM2-SpCas9 is the therapeutic vector. CMV promoters are notorious for a short half life. The control should be rAAV1-ICAM2-GFP.

The GFP was used as an indicator of transduction efficiency because there had been no report on the efficiency of AAV transduction of ECs in these two models. Initially, we used rAAV1-pICAM2-GFP as control for demonstrating transduction efficiency, but the GFP signal was too weak so we switched to rAAV1- CMV-GFP to demonstrate transduction efficiency. Notably, the neither CMV nor pICAM2 promoter would affect the expression of sgRNA in the SpGuide.

Other comments:

Figure 2. It appears that transfection occurs in large blood vessels not capillaries. Optimally, the therapy should target capillaries not large blood vessels. Are capillaries transfected and the photo is not good enough to see that or is there truly no transfection of capillaries?

The rAAV1 preferentially transduced ECs of pathological vessels. As shown in the new supplemental figure 2, EC of pathological capillaries were also transfected.

Reviewer #3

The results are interesting and exciting but because this is one of the first papers using gene editing to manipulate an ocular disease model, it would be helpful to have some additional information:

1) There is no long term follow-up data, presumably because the animal models are

acute ones. But it would be helpful to know whether permanent expression of the transgenes resulted in any toxicity, particularly as they probably have not reached peak expression by the time that the readouts were made (5-7 days after injection). In particular, does production of the SpCas9 cause any toxicity/immune response?

As recommended, we injected the dual AAVs intravitreally into P12 pups to determine if they would cause any toxicity or immune responses after four weeks. New results presented in new supplemental figure 5 showed that there was no toxicity to the retina structure and function, and vascular function examined by optical coherence tomography (OCT), electroretinography (ERG), fundus angiography (FA), and retinal whole mounts staining with murine-specific EC marker isolectin 4 (IB4)-Alexa 594 (red). The injected eyes did not show any obvious immune response (data not shown).

2) Does the system interfere with normal blood vessel growth? I presume that the timepoint for delivery in the ROP model (P12) was selected to minimize this possibility. Should this be discussed?

To address this question, we intravitreally injected the dual AAVs into P12 pups. The results in the new supplemental Figure 5 showed that there was no significant difference in the fundus photo of FA between sham-control and AAV injection. We have added this point in the Discussion section.

3) Interestingly, in the laser photocoagulation model, intravitreal injection of the reagents ameliorates the outer retinal phenotype. Is there evidence that AAV1 penetrates across the mouse retina from the vitreal aspect or could it penetrate through the laser-damaged regions of the retina?

As showed in the new supplemental Figure 4, AAV1 is able to penetrate through the laser-damaged regions of the retina.

4) How could this approach be extrapolated to treat humans? For choroidal neovascularization, the reagents would presumably have to be injected into the subretinal space in direct contact with the aberrant blood vessels. Would the reagents be delivered after the disease process has initiated or before (as with ROP mouse)? If the reagents target mainly pathologic vessels, that may be the case. Also, do you expect that the sequence used to target the mouse VEGFR2 sequence would be the same target used for human disease (ie, are they homologous?)? What steps would be necessary to test a sequence that could be applied to humans.

To address these topics, we have added comment in the discussion. To test whether this sequence could be applied to humans, we would need to examine its efficiency of editing VEGFR2 in human retinal microvascular endothelial cells (HRECs). For this, we would have to clone the homologous sgRNA (mK22) from human VEGFR2 into this AAV SpGuide vector and test if the K12 (human homologue to mK22) was able to deplete expression of VEGFR2 in HRECs. Next, we would have to examine whether genome editing of VEGFR2 could block VEGF-induced migration and tube formation by HRECs in vitro.

5) Relevant to the discussion about need for additional therapies, for humans who do not respond to the available therapies for ocular neovascularization, is there any data suggesting that there are other pathways/targets (besides VEGF) that could be awry?

Besides VEGF matrix metalloproteinase (MMP) is also a potential therapeutic target; we have added this comment in this revision.

Other:

1) Results: "Transduction of ECs...", ~3 sentences from end of section: brush' membrane? I assume this is Bruch's.

We have corrected brush' to Bruch's in this revision.

REVIEWERS' COMMENTS:

Reviewer #1 (Remarks to the Author):

In this revised manuscript the authors have significantly improved their manuscript. In particular, the authors have measured off-target effects by deep sequencing and quantified the proportion of GFP positive ECs. All of my concerns have been adequately addressed.

Reviewer #3 (Remarks to the Author):

The authors have addressed my initial concerns satisfactorily.